# A Combined Method of Seismic Monitoring and Transient Electromagnetic Detection for the Evaluation of Hydraulic Fracturing Effect in Coal Burst Prevention

**DOI:** 10.3390/s24061771

**Published:** 2024-03-09

**Authors:** Jiang Bian, Aixin Liu, Shuo Yang, Qiang Lu, Bo Jia, Fuhong Li, Xingen Ma, Siyuan Gong, Wu Cai

**Affiliations:** 1School of Mines, China University of Mining and Technology, Xuzhou 221116, China; ts21020216p21@cumt.edu.cn (J.B.);; 2State Key Laboratory for Fine Exploration and Intelligent Development of Coal Resources, China University of Mining and Technology, Xuzhou 221116, China; 3Hetaoyu Coal Mine, Huaneng Qingyang Coal Power Co., Ltd., Qingyang 745300, China; 4Huaneng Coal Technology Research Co., Ltd., Beijing 100070, China

**Keywords:** coal burst prevention, hydraulic fracturing, microseismic monitoring, transient electromagnetic detection, fracturing effect evaluation

## Abstract

In order to mitigate the risk of roof-dominated coal burst in underground coal mining, horizontal long borehole staged hydraulic fracturing technology has been prevailingly employed to facilitate the weakening treatment of the hard roof in advance. Such weakening effect, however, can hardly be evaluated, which leads to a lack of a basis in which to design the schemes and parameters of hydraulic fracturing. In this study, a combined underground–ground integrated microseismic monitoring and transient electromagnetic detection method was utilized to carry out simultaneous evaluations of the seismic responses to each staged fracturing and the apparent resistivity changes before and after all finished fracturing. On this basis, the comparable and applicable fracturing effects on coal burst prevention were evaluated and validated by the distribution of microseismic events and their energy magnitude during the mining process. Results show that the observed mining-induced seismic events are consistent with the evaluation results obtained from the combined seismic-electromagnetic detection method. However, there is a limited reduction effect on resistivity near the fractured section that induces far-field seismic events. Mining-induced seismic events are concentrated primarily within specific areas, while microseismic events in the fractured area exhibit high frequency but low energy overall. This study validates the rationality of combined seismic-electromagnetic detection results and provides valuable insights for optimizing fracturing construction schemes as well as comprehensively evaluating outcomes associated with underground directional long borehole staged hydraulic fracturing.

## 1. Introduction

Hydraulic fracturing, especially within the horizontal long borehole staged fracturing technology (L-shape hydraulic fracturing), was initially employed in the field of oil and gas extraction. By injecting fluid into the borehole through high-pressure pumps, fractures are stimulated in the rock when the fluid pressure exceeds its strength. Such fractures serve as channels to improve the extraction efficiency of oil, gas, and geothermal resources [1]. Since the 1990s, hydraulic fracturing technology has been gradually applied in coalbed methane gas management and coal burst prevention in coal mines [2,3]. Its objectives include enhancing coal seam permeability [4,5], increasing the gas extraction rate [6], weakening hard roofs [7,8], reducing the initial pressure step distance [9,10], and minimizing coal burst tendency [11,12]. In recent years, the application of directional hydraulic fracturing in coal burst prevention has been demonstrated in a few coal mines. For instance, Su [13] took Hejiata Coal Mine in Shenfu Coalfield as the application object, and verified that after hydraulic fracturing pre-split roof, the roof can be stratified and caved successively, and the initial strength of the old roof can be weakened. Liu et al. [14] adopted the hydraulic fracturing construction technology of ultra-thick hard roof, judged the degree of crack development through drilling and peeking, and verified the pressure relief effect according to the peak working resistance and load increase coefficient of the support. Pang et al. [15] designed the fracturing scheme based on the key layer theory and Hubbert–Willis non-permeable formation rupture pressure calculation formula, and verified the good effect of directional hydraulic fracturing with long horizontal holes by using the periodic pressure step distance and advanced support pressure.

Currently, the monitoring techniques for hydraulic fracturing prevention and control technology primarily encompass borehole imaging, well temperature logging, transient electromagnetic detection [16], direct current detection, microseismic monitoring [17,18], and construction parameter analysis. Men et al. [19] adopted ARAMIS/E microseismic monitoring and ARES-5 ground sound monitoring to conduct the joint monitoring of hydraulic fracturing of ultra-long boreholes in the roof of ultra-thick coal seam. The results show that the fracture of the rock layer is mainly in a small area and has a high frequency during fracturing, which effectively reduces the energy accumulation of the overlying rock layer on the roof. Kim J et al. [20] used the crosswell electromagnetic (EM) geophysical methods to detect the fracture propagation induced by hydraulic fracturing; the resulting conductivity images provided a clearer understanding of the fractured reservoir, proving the EM can serve as an effective monitoring tool for monitoring the injected fluids during hydraulic fracturing. Kang et al. [21] used ground three-component microseismic monitoring to analyze the distribution of a fracture network, and used a mine pressure monitoring and warning platform, borehole stress gauge, underground microseismic monitoring, etc., to form a hydraulic fracturing analysis system for directional drilling in coal mines. Zhang et al. [22] analyzed the fracture area and fracturing influence range of coal and rock mass after fracturing by comparing the numerical simulation results with the transient electromagnetic detection. Peng et al. [23] used a vibration wave CT monitoring method and direct current method to reflect the extension degree of fractures around fracture holes and the connectivity and extension degree of primary fractures, and built a multi-variable and multi-scale detection method for the influence range of hydraulic fracturing.

In this study, the engineering background was provided by the fully mechanized top coal caving face longwall panel 1802 in Hetaoyu Coal Mine, where the application and analysis of combined seismic-electromagnetic monitoring technology for roof hydraulic fracturing effect evaluation were conducted. Based on the results obtained from simultaneous microseismic monitoring during fracturing and transient electromagnetic detection measuring apparent resistivity changes before and after fracturing, the effectiveness of the fracturing project was evaluated to establish a foundation for enhancing and optimizing the fracturing scheme. Finally, the effectiveness of this method is further verified by analyzing the pressure relief effect of each region through the microseismic events monitored during the mining process of the working face.

## 2. Project Overview

### 2.1. Geological Survey of the Mine

The Hetaoyu Mine is situated at Gansu Province, China, spanning approximately 14.728 km from east to west and 12.438 km from north to south, the mine covers an expansive area of about 183.189 km^2^. There are 3 coal seams (No. 2, No. 5, and No. 8) to be mined, in which the No. 8 coal seam was designed as the primary mining layer [24]. There are two fully mechanized caving faces in production at Hetaoyu mine: 2804 and 1802. The coal seam’s roof and floor rocks exhibit low solidity, weak rock strength, uneven local bump conditions, limited fissure development, expansion of both the roof and floor when encountering water, frequent floor occurrences, a high risk of roof collapse, as well as a significant presence of gas, making it a high-risk mine for explosion due to coal dust accumulation [25]. The three-dimensional geological survey of the Hetaoyu Mine is shown in Figure 1, within the location of drilling field 1 included.

### 2.2. Directional Long Borehole Staged Hydraulic Fracturing

In the context of fully mechanized mining of thick coal seams with hard roofs, the directional long borehole staged hydraulic fracturing is a technical measure for the regional advanced transformation of roof stress distribution.

The construction area of drilling field 1 of Hetaoyu Coal Mine studied in this paper is located in the initial mining area of the 1802 working face, and the mining of the working face passes through the square special structure area and the S5 anticline area, and the impact risk is high. In order to investigate the hydraulic fracturing relief effect of the key layer of overlying rock, three boreholes, No. 1, No. 2, and No. 3, were designed by directional drilling technology. According to the borehole column shape, the occurrence general situation of the strata above the coal seam was determined, and the fracturing plan of the key layer was designed, and the construction effect of the staged fracturing is shown in Figure 2. The hole opening position of drilling field 1 is about 530 m away from the open cut-off the working face 1802. All holes are drilled in single-hole mode. Table 1 describes the drilling parameters.

In the process of fracturing engineering implementation, the number of fracturing stages and the spacing of each fracturing stage should be adjusted according to the exposure of rock strata during drilling. The actual fracturing sequence of the first drilling field is borehole No. 3 (frac stages 1–9) → borehole No. 1 → borehole No. 2 → borehole No. 3 (frac stages 10–14). The hole depth of borehole No. 1 is about 531 m, and there are 12 stages of fracturing. The hole depth of borehole No. 2 is about 522 m and there are 14 stages of fracturing. Borehole No. 3 has a depth of approximately 573 m and a total of 14 stages of fracturing. Figure 3 shows the distribution of pressure fractures. Table 2 lists the length data of each fracturing stage.

## 3. Combined Seismic-Electromagnetic Detection Method

### 3.1. Underground–Ground Integrated Microseismic Monitoring System

The joint monitoring system utilized in this study primarily comprises the robust well vibration monitoring system and the subterranean SOS microseismic monitoring system. Building upon the pre-existing SOS microseismic monitoring system at Hetaoyu Coal Mine, this system meticulously designs and arranges a ground-based three-component strong seismograph to achieve the synchronized transmission of vibration data via the 4G network. Consequently, it establishes an all-encompassing three-dimensional envelope of the monitored area, enhancing the positioning accuracy of the microseismic monitoring system while reducing positioning errors in the z direction for individual underground pickups. This refined approach enables the more effective surveillance of vibration locations resulting from overlying rock fractures during hydraulic fracturing operations and facilitates the accurate assessment of fracture development levels [26,27]. The microseismic location diagram of the combined well and ground system is shown in Figure 4 below, and the microseismic parameters of underground and ground sensors are shown in Table 3.

For the purpose of achieving high accuracy in the source location, the selection of the longitudinal wave (P wave), which is more discernible, is typically favored for localization, as shown in Figure 5. Compared to other waves, the determination error of the first arrival time of the P wave is smaller and its positioning accuracy is higher. Our system uses the STA/LTA method to automatically identify the microseismic signal. Due to the low signal-to-noise ratio of the mining area, it is necessary to manually adjust the initial arrival time of the P wave, and the theoretical and actual calibration time of each sensor is marked. The deviation is controlled within ±20 ms. For signals with noise interference, the FFT algorithm within the low-pass filtering method is used for filtering and calibration. In practical applications, it is often assumed that the vibration wave propagates in a uniform and isotropic medium, making it very challenging to determine the propagation time using any propagation velocity. This assumption implies that the P wave maintains a constant velocity in all propagation directions. The formula below represents the minimum time taken for the vibration signal to travel from its source to the pickup location.
(1)$$t_{i}-t_{0}=\frac{\sqrt{{\left(x_{0}-x_{i}\right)}^{2}+{\left(y_{0}-y_{i}\right)}^{2}+{\left(z_{0}-z_{i}\right)}^{2}}}{v\left(x_{o},y_{0},z_{0}\right)}$$

In the formula, $x_{0}$, $y_{0}$, $z_{0}$ are the source coordinates; $t_{0}$ is the time of onset of seismic source;$\;x_{i}$, $y_{i}$, $z_{i}$ are the coordinates of the seismic geophone;$\;t_{i}$ is the arrival time of the P wave at the seismic geophone;$\;v\left(x_{0},y_{0},z_{0}\right)$ is the velocity of P wave propagation in medium.

According to the fracturing area of drilling field 1 in the 1802 working face of Hetaoyu Coal Mine, considering the actual equipment installation and optimization of the microseismic monitoring effect during fracturing, a layout scheme comprising 13 sensors serving drilling field 1 was devised, as shown in Figure 6. Among them, a total of 7 SOS sensors were assembled downhole at the two roadways of 1802 working face; U19 is a large-range sensor, and its installation position is only 0.5 m away from the U18 sensor, which is used for the energy calculation of the large vibration. The rest are small-range sensors, which are used for the precise positioning of the microseismic source. And a total of 6 ET-GSY strong seismograph stations were assembled on the ground, which finally form an underground–ground envelope monitoring system for the fracturing area of drilling field 1.

By using the seismic wave passive CT inversion technology, the change in the impact dangerous area of drilling field 1 in 1802 working face before and after fracturing can be identified and analyzed. A microseismic monitoring system is utilized to calibrate the triggering time of downhole vibrations and determine the spatial distance between rupture positions and sensors. When vibration waves propagate through coal and rock masses, they induce changes in wave velocity due to stress effects [28].

The abnormal value of wave velocity can better reflect the stress distribution state of the detection area, thus providing a basis for the division of impact hazard area and the implementation of pressure relief prevention and control measures. The higher the abnormal value the wave velocity is, the stronger the stress concentration characteristics and the larger the concentration probability is, and the stronger the impact hazard is. The calculation method for determining the abnormal value of wave velocity is presented in Formula (2), while the correlation between the abnormal value of wave velocity and the probability of stress concentration is illustrated in Table 4.
(2)$$A_{n}=\frac{v_{p}-v_{p}^{a}}{v_{p}^{a}}$$

In the formula, $v_{p}$ is the P-wave velocity value at a point in the inversion region; $v_{p}^{a}$ is the average of the model wave velocity.

### 3.2. Transient Electromagnetic Detection

The principle of mine transient electromagnetic detection is the principle of electromagnetic induction. A transmitting coil with a certain current is set up outside the detection area by using an ungrounded return line, and an induced current is generated in the conductive rock orebody around the detection area. The current establishes a stable magnetic field in the space around the transmitting coil. At any initial moment, when the original electromagnetic field is suddenly closed or changed, the underground free electrons cannot stop moving immediately because of inertia, so they will continue to produce secondary electromagnetic fields, which is the so-called “transient” process. By measuring the intensity and direction of the transient electromagnetic field, the position, shape, size, depth, and electrical parameters of the anomalous body can be obtained [29]. Figure 7 shows the detector host.

The primary objective of this study is to identify the development and expansion of hydrofractures subsequent to fracturing the roof of the working face. In rock masses characterized by good integrity and compactness, their resistivity tends to be relatively high. However, when fracturing fluid penetrates through the rock mass, leading to fracture generation and expansion, the resistivity in these fractured regions decreases due to fluid influx [30].

The design detection route is shown in Figure 8. Prior to and following fracturing in the drilling field 1, detection commenced from the 1802 cut hole, with a detection point every 10 m along the air return lane and transportation lane. To comprehensively monitor fissure development within the stratum, three detection angles were designated for each point of measurement, utilizing 2 m × 2 m electromagnetic coils to measure angles (15°, 45°, and 60°) relative to the coal wall of the working face. The detection schematic diagram is shown in Figure 9, and the field measurement is shown in Figure 10.

## 4. Application Results and Fracturing Effect Analysis

### 4.1. Analysis of Fracturing Construction Parameters

The accumulated fracturing fluid in borehole No. 1 is 458.9 m^3^, with the maximum construction pressure ranging from 22.4 MPa to 34.9 MPa and the maximum pressure drop ranging from 4.9 MPa to 12.8 MPa. The accumulated fracturing fluid in borehole No. 2 is 660.05 m^3^, with the maximum construction pressure ranging from 12.40 MPa to 33.6 MPa and the maximum pressure drop ranging from 0.7 MPa to 16 MPa. The accumulated fracturing fluid in borehole No. 3 is 581.75 m^3^, with the maximum construction pressure ranging from 16.1 MPa to 29.8 MPa and the maximum pressure drop ranging from 2.2 MPa to 12.9 MPa. Figure 11 is the pressure–flow curve during the water injection period of the fracturing stage. It can be reflected from the curve that the pressure of the fracturing stage rises to a critical value and then a sudden pressure drop occurs. This is due to the fact that the fracturing fluid breaks through and produces fractures. This critical value is the fracture initiation pressure, and then with the continuous injection of the fracturing fluid, the fractures are expanded, and the pressure and flow are in a serrated stable state.

The initiation pressure and the ratio of injection pressure to injection flow in each fracturing stage of the borehole are shown in Figure 12. The pressure changes of different fracturing stages are not regular in the region, indicating that the strength, integrity, fracture development, permeability, and other factors of the rock layer are not stable and vary greatly. In general, the initiation pressure (as shown in Figure 12a) can serve as an indicator of the maximum pressure exerted by the fracturing fluid on the packer during its passage through the rock layer. A higher value signifies a stronger rock layer in this particular area. Furthermore, the ratio of injection pressure to injection flow (as shown in Figure 12b) provides insights into the extent of rock fracture expansion within the scope of hydraulic fracturing [31,32]. However, it is difficult to accurately verify and evaluate the effect of fracturing through simple construction parameters. Therefore, a reliable geophysical detection method is needed more to obtain the data changes during and before fracturing, and to effectively evaluate the effect of fracturing in real time [33].

### 4.2. Analysis of Microseismic Monitoring Results

During the fracturing process of drilling field 1, the joint well-ground microseismic monitoring system provided real-time monitoring and successfully conducted an effective waveform analysis. In this regard, the joint well-ground monitoring system of borehole No. 1 recorded a total of 115 microseismic events in working face 1802, with a maximum microseismic energy of 19,522.5 J. The focal distribution is shown in Figure 13a.

A total of 311 microseismic events were recorded by the joint monitoring system of borehole No. 2 at the working face of 1802, with the maximum microseismic energy of 7755.6 J. The focal distribution is shown in Figure 13b.

A total of 270 microseismic events were recorded by the joint monitoring system of borehole No. 3 at the working face of 1802, with the maximum microseismic energy of 38,211.8 J. The focal distribution is shown in Figure 13c.

After cross-referencing the daily production operation of the mine, it was determined that the vibration events in Area A during the fracturing of the three boreholes were attributed to excavation activities related to the 1802 special drainage roadway. Similarly, in Area B, where the No. 2 borehole was being fractured, pipeline installation operations were underway. Hence, it can be inferred that both the vibration events in Area A and No. 2 borehole’s Area B are unrelated to hydraulic fracturing activities. Further examination revealed that the microseismic event in No. 2 borehole’s Area C resulted from blasting operations and not hydraulic fracturing-induced phenomena.

Figure 14 shows all frac microseismic events generated by borehole No. 1 and the corresponding frac stages. Among them, five microseismic events were induced during the fifth stage of fracturing, with very small energy levels, and the maximum energy recorded was only 54 J. Based on the coordinates and spatial location of these events, it can be observed that the vibrations occurred above the roof of the 1802 return air lane but at a distance greater than 60 m from the fracturing site. Furthermore, these five vibrations took place during a stable pressure and flow stage rather than when there was a significant reduction in water pressure. Therefore, it is concluded that these microseismic events were not directly caused by fracture propagation during water injection. Instead, two possibilities exist.

Firstly, after hydraulic fracturing altered the long-distance formation stress field, weak surfaces and natural fractures experienced breakage due to changes or disturbances in this stress field. The fact that these vibrations occurred at locations where roof leakage was detected suggests communication between fractures induced by vibration and those formed due to water pressure, thus forming a pathway for water flow [34,35].

Secondly, it is speculated that reduced cohesion resulting from water entering pre-existing fractures (which are relatively well developed at leakage sites) led to decreased effective normal stress (increased pore pressure), thereby inducing the fracture slip instability [36,37].

Subsequently, as shown in Figure 14, the occurrence of a mine earthquake event with an energy of 1.9 × 10^4^ J outside the return air lane shortly after the seventh stage fracturing has been fully demonstrated by monitoring results, showing that it is challenging to induce high-energy microseismic events at the location of rock strata where hydraulic fracturing takes place. Therefore, this event is also attributed to far-field rock strata instability caused by hydraulic pressure. Additionally, it indicates significant stress conduction between hydraulic pressure and fractures during the hydraulic fracturing process, which cannot be achieved solely through small-scale fractures [38].

Furthermore, it has been confirmed that no operation was conducted during the fracturing of the borehole No. 3 B area. However, a microseismic event was induced during the fifth stage fracturing with very low energy (only 78 J), as shown in Figure 15. Considering the coordinates and spatial location of this microseismic event, vibrations occurred over a distance exceeding 60 m from the fracturing interval and took place during flow reduction and pressure reduction stages. Hence, it can be determined that this microseismic event is not directly caused by crack propagation in the water injection process but it is rather due to weak surface ruptures and natural fractures being disturbed or altered by changes in the long-distance formation stress field induced by hydraulic fracturing activities [39,40].

As shown in Figure 16, the microseismic waveforms before and after fracturing were marked to determine the source location, and the analysis focused on the 220 m mark level with the highest ray coverage density. The marked area in the figure is the tunneling operation of the outer drainage roadway of the transportation lane. These operations induce redistribution of the original stress field in the roof, leading to stress concentration. Apart from this tunneling area, a comparison of imaging data reveals a stress concentration zone with abnormally high wave velocity values extending approximately 400 m along the working face from the middle of drilling field 1 to the side of the transport lane prior to the fracturing construction. However, after the fracturing construction, both coverage area and abnormal wave velocity values within this zone are significantly reduced (as shown in Figure 16c). Furthermore, there is an increased extent of low abnormal wave velocity values observed through No. 1 and No. 2 boreholes, indicating that hydraulic fracturing has expanded fracture fissures in alignment with the maximum principal stress direction resulting in fractures within hard rock layers of roof strata [41,42].

For areas where abnormal wave velocity values remain higher than 0.25 even after fracturing (especially within intermediate regions between borehole No. 3 and the transport lane), additional pressure relief measures such as coal blasting should be reinforced to prevent potential rock burst incidents during the mining operation.

### 4.3. Analysis of Transient Electromagnetic Detection Results

According to the transient electromagnetic detection scheme of drilling field 1 formulated in Section 2.2, 53 points (530 m) were detected from the cutting hole of the working face 1802, each of which included three angles of 15°, 45°, and 60°. Two probes were conducted before and after fracturing. Figure 17 shows the apparent resistivity distribution of drilling field 1 at three detection angles. The apparent resistivity before fracturing is subtracted from the apparent resistivity after fracturing, which can reflect the fracture expansion after fracturing the high-pressure water injection, if the difference between the two is greater than 0, that is, the apparent resistivity decreases, indicating that the fracture expands to this area after fracturing [43].

The three-dimensional distribution of resistivity before and after fracturing along the horizontal and vertical directions of the borehole is illustrated in Figure 18 [44]. It can be observed that, apart from the influence of the advance bracket within a 150 m range near the cut hole, there has been a significant reduction in resistivity throughout the entire drilling field 1 in the horizontal direction. In terms of vertical direction, borehole No. 1 exhibits an overall superior fracturing effect with reduced resistivity near all stages of fracturing; particularly notable is the substantial decrease in resistivity at stages 5–12. On the other hand, while not evident during stages 1–4, borehole No. 3 demonstrates a remarkable reduction in resistivity during stages 5–14, indicating a significant fracturing effect [45,46].

By comparing the injection pressure–injection flow ratio distribution cloud chart and the CT inversion results of the vibration wave after fracturing, specifically examining the distribution of abnormal wave velocity values in Figure 16b, it can be observed that these values correspond to the spatial distribution area of apparent resistivity. Moreover, a positive correlation between them is evident; requisite fracturing is insufficient in areas with high wave velocity abnormal values and high apparent resistivity, thereby indicating that the integration of these two detection methods can effectively verify and analyze the effectiveness of fracturing.

### 4.4. Fracture Radius Analysis

According to the combined seismic-electromagnetic detection results, the fracturing radius is determined through analysis, specifically by calculating the distance of microseismic sources induced by fracturing and assessing the range of apparent resistivity reduction post-fracturing. This enables an evaluation of the efficacy of hydraulic fracturing at each stage [47,48].

As shown in Figure 19, the fracturing radius $r_{2}$ is satisfied as follows:(3)$$r_{2}>L-r_{1}$$

In the formula, $r_{1}$ is the focal rupture radius, and *L* is the vertical distance between the fracturing location and the 1802 return air lane, which is 66 m.

Therefore, according to Formula (3), the fracturing radius can be estimated as long as the rupture radius of the source is calculated [49].

Generally, the larger the energy, the larger the rupture radius. Therefore, a microseismic signal with the largest energy induced by disturbance during the fracturing of the fifth section of borehole No. 1 is calculated to calculate the rupture radius of the source. Brune’s model is selected as the calculation model, and the calculation formula is as follows [50]:(4)$$r=\frac{K\beta }{2\pi f_{c}}$$

K is the constant selected based on Brune’s model, P wave is 2.01, S wave is 1.32; β is the vibration wave velocity, P wave is 4670 m/s, S wave is 2696 m/s; $f_{c}$ is the corner frequency, expressed in Hz. As shown in the signal spectrum in Figure 20, the calculated average corner frequency is 65 Hz, and the calculated corner frequency, P-wave velocity and S-wave velocity of each channel are inserted into Formula (4), and the calculated average source radius is 16.1 m.

Therefore, by substituting $r_{1}$ = 16.1 and *L* = 66 into Formula (3), the calculation shows that the fracturing radius $r_{2}$ is at least greater than 49.9 m.
$$r_{2}>L-r_{1}=66-16.1=49.9\;m$$

The overall distribution of apparent resistivity detected by transient electromagnetic detection along the horizontal section of the borehole is illustrated in Figure 21. In proximity to the cut-off hole, the apparent resistivity detection value is influenced by the advance bracket and deviates from its actual value. It can be observed that fracturing has led to a certain reduction in apparent resistivity near the borehole, with some areas experiencing a reduction range exceeding 2 Ω·m, indicating effective diffusion of fracturing fluid.

Utilizing software (Voxler 4) analysis, we have determined that the radius encompassing areas with the rate of change of apparent resistivity greater than 0.5 (as shown in Figure 21c) ranges from 29.8–54.7 m, representing an effective zone for hydraulic fracturing operations. By integrating this information with seismic source rupture calculations, our comprehensive analysis suggests that the effective fracturing radius falls within a range of 29.8–54.7 m [51].

### 4.5. Fracturing Effect Evaluation

The pressure relief effect of roof hydraulic fracturing can be analyzed and evaluated based on microseismic data collected in the fractured area during face mining operations. Currently, there are two operational working faces at Hetaoyu Coal Mine: the 2804 fully mechanized caving working face and the 1802 fully mechanized caving working face. Among these, only the 1802 working face has implemented segmented hydraulic fracturing using directional long boreholes. By analyzing statistical data on pushing and mining advancement, it is observed that both working faces maintain a relatively stable mining speed during the initial stage of mining. To comparatively analyze daily frequency and total energy of microseismic events, all monitored mining microseismic events from the initial stage (first 200 m of advancement and first 50 days of stable pushing and mining) for both working faces were selected for analysis (as shown in Figure 22). A total of 1864 microseismic events with a total energy of 4,354,002.878 J were recorded in the working face 2804 at 200 m before the initial mining. A total of 622 microseismic events were recorded at the 1802 face, with a total energy of 1,495,330.004 J. A total of 995 microseismic events were recorded in the first 50 days of mining at the 2804 working face, with a total energy of 2,126,645.502 J. A total of 611 microseismic events were recorded at the 1802 face, with a total energy of 1,502,743.076 J.

By comparing the daily microseismic frequency and energy statistics of the two working faces depicted in Figure 22, it can be observed that the 1802 working face, which implements roof hydraulic fracturing, exhibits a general reduction in the number of microseismic events and total energy during the initial mining stage. Moreover, only a few high-energy events are induced. The total energy associated with each day’s microseismic activity is directly proportional to its vibration frequency. In Figure 22a,c, as the 1802 working face advances inward from the cut hole, there is an increase in microseismic events after passing through the hydraulic fracturing area; however, these events exhibit relatively low average energy levels, indicating a distribution characterized by “high frequency and low energy”. This observation reflects the effective pressure relief achieved through directional long borehole segmented hydraulic fracturing [52].

The spatial evolution of microseismic frequency and energy in the initial mining stage of 1802 and 2804 working faces is illustrated in Figure 23. For the 1802 working face with fracturing, microseismic events generated during mining are primarily concentrated on both sides of the working face. Moreover, compared to the non-fractured working face, there is a significant reduction in vibration frequency and energy, indicating that the segmented hydraulic fracturing construction of directional long boreholes has effectively alleviated pressure.

By considering the position of hydraulic fracturing boreholes in drilling field 1, it can be observed that the stress concentration phenomena occur near the fifth fracturing segment of borehole No. 1 and the fifth fracturing segment of borehole No. 3. These areas experience high-frequency vibrations with a substantial total energy release during mining operations, which aligns with well-ground joint microseismic monitoring results during fracturing activities as well as CT inversion results for seismic wave analysis (Section 4.2) and resistivity distribution along the borehole after fracturing (Section 4.4). This suggests that these specific regions exhibit high-strength overburden rock layers where extensive fracture expansion occurs during hydraulic fracturing processes, leading to far-field stress disturbances induced by such activities. Notably, unlike other areas where post-fracture resistivity reduction is more pronounced, this particular region demonstrates less obvious changes in resistivity after fracturing while still maintaining an elevated stress intensity within its overburden rock layer, consequently resulting in concentrated energy release phenomena during subsequent mining operations, thus validating the effectiveness of employing combined seismic-electromagnetic detection methodologies adopted within this study.

## 5. Conclusions

This paper adopts a combined seismic-electromagnetic detection method that combines an underground–ground microseismic monitoring with mining transient electromagnetic detection to quantify and analyze the spatial distribution of microseismic events and apparent resistivity during hydraulic fracturing in the hard roof of the 1802 working face at the Hetaoyu Coal Mine. The subsequent findings are as follows:Hydraulic fracturing poses challenges in directly inducing microseismic events and generates low levels of vibration energy. However, the pressure generated by water injection and stress conduction resulting from fracture expansion can indirectly induce far-field vibrations. This is particularly evident in areas with naturally weak surfaces, highly fractured rock masses, or concentrated stress zones featuring faults, anticlines, and other unique geological structures. After fracturing, these rock layers may experience instability slips accompanied by elevated abnormal wave velocity values.The transient electromagnetic detection of the apparent resistivity in different layers of coal and rock mass reveals that the roof rock layer exhibits a lower apparent resistivity after fracturing, indicating successful penetration of fracturing fluid into generated cracks and overall good fracturing effectiveness. However, certain areas still exhibit high apparent resistivity post-fracturing, aligning with the distribution of hydraulic flow and microseismic sources, suggesting suboptimal fracturing outcomes.According to the microseismic monitoring data during the mining period in the fractured area, a significant reduction in both frequency and number of high-energy mining vibration events was observed across the entire working face after fracturing, as compared to the non-fractured working face. Furthermore, these findings align with those obtained from combined seismic-electromagnetic detection, confirming that the concentrated vibration areas are consistent. Additionally, it was found that microseismic events occurring in the regional mining post-fracturing exhibit a distribution characterized by “high frequency and low energy”. These results indicate that staged fracturing technology effectively alleviates stress concentration phenomena associated with hard roof strata while reducing the potential for rock burst disasters on the working face.The combined seismic-electromagnetic detection accurately reflects the impact of hydraulic fracturing on the overlying strata. The results obtained from the combined approach of joint well-ground microseismic monitoring and transient electromagnetic detection are in correspondence. This combination strengthens the purpose of effect verification and facilitates analysis of potential shortcomings in fracturing operations. In future studies, it is imperative to precisely determine an optimal fracturing layer, ensure effective fracturing within critical rock layers, promote fracture development, and implement robust measures to mitigate erosion and relieve excessive pressure in areas where satisfactory fracturing outcomes have not been achieved.

## Figures and Tables

**Figure 1 sensors-24-01771-f001:**
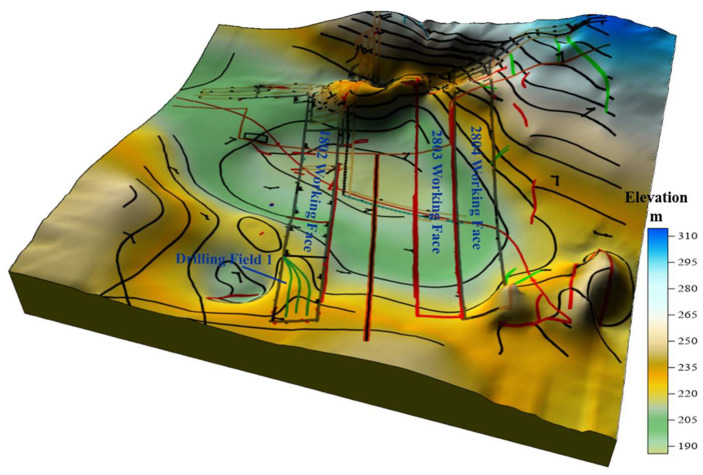
Three dimensional geological map of Hetaoyu mine.

**Figure 2 sensors-24-01771-f002:**
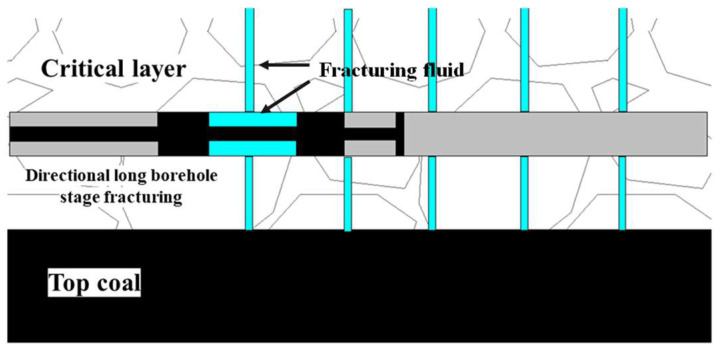
Sectional fracturing construction effect of directional long drilling.

**Figure 3 sensors-24-01771-f003:**
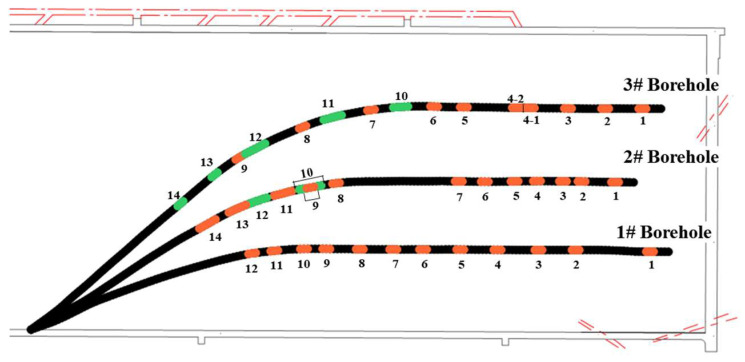
Distribution of fracturing stages of each borehole in drilling field 1.

**Figure 4 sensors-24-01771-f004:**
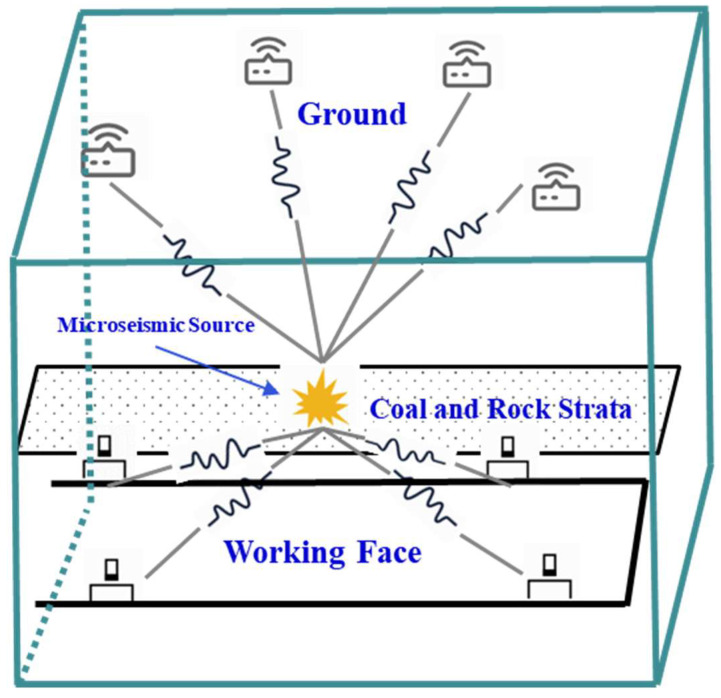
Schematic diagram of underground–ground integrated microseismic monitoring system.

**Figure 5 sensors-24-01771-f005:**
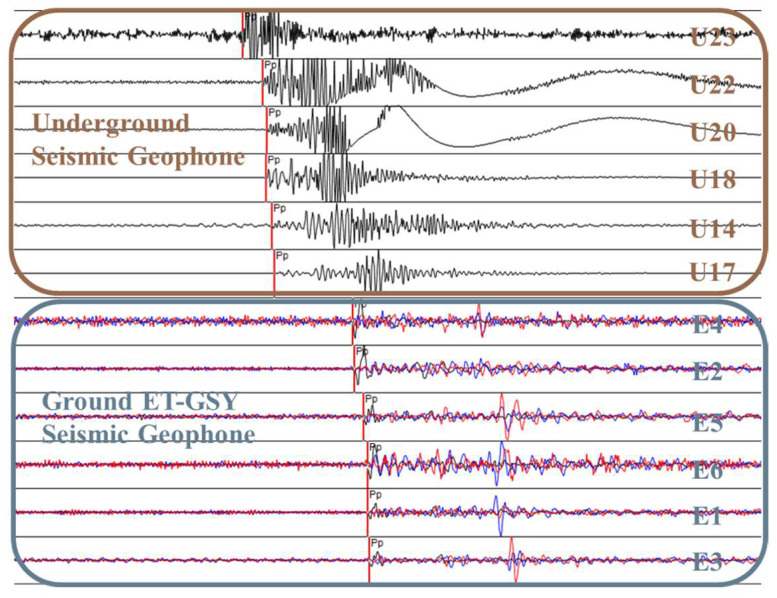
Schematic diagram of source P wave marking location.

**Figure 6 sensors-24-01771-f006:**
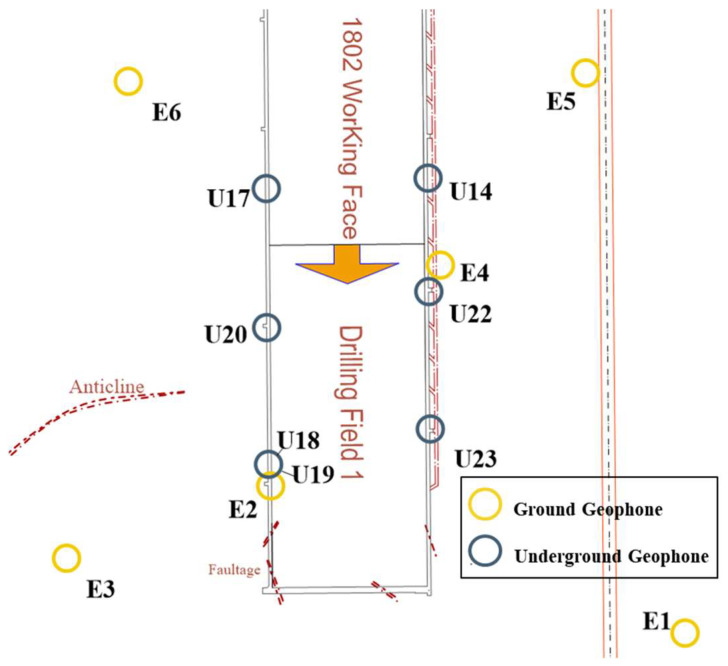
Layout scheme of the underground–ground integrated microseismic sensors.

**Figure 7 sensors-24-01771-f007:**
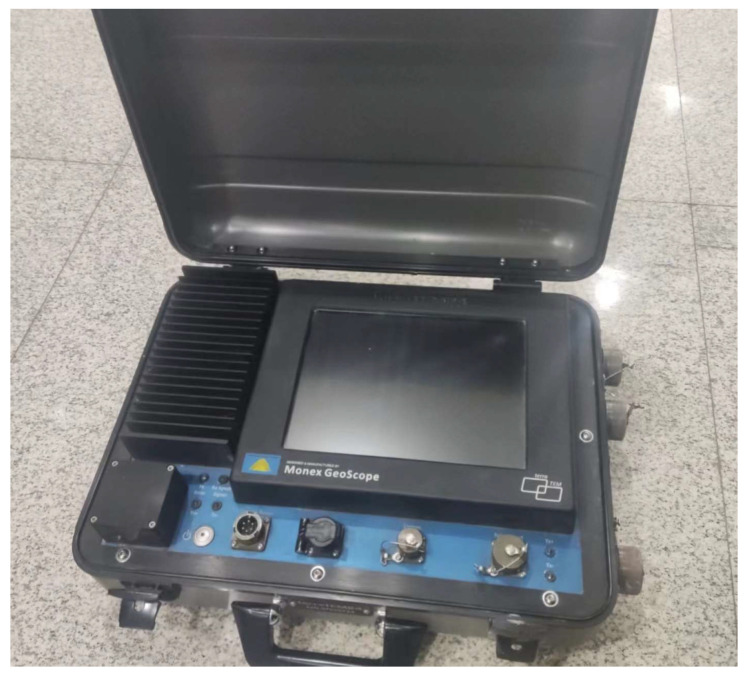
Host of the transient electromagnetic meter.

**Figure 8 sensors-24-01771-f008:**
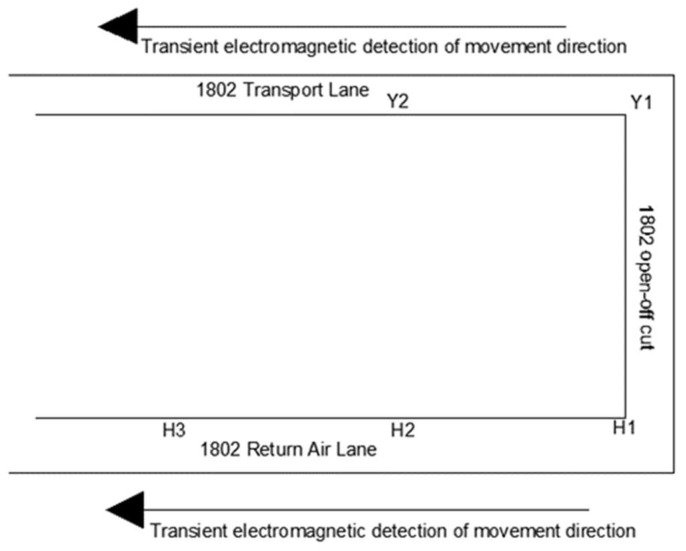
Transient electromagnetic detection direction.

**Figure 9 sensors-24-01771-f009:**
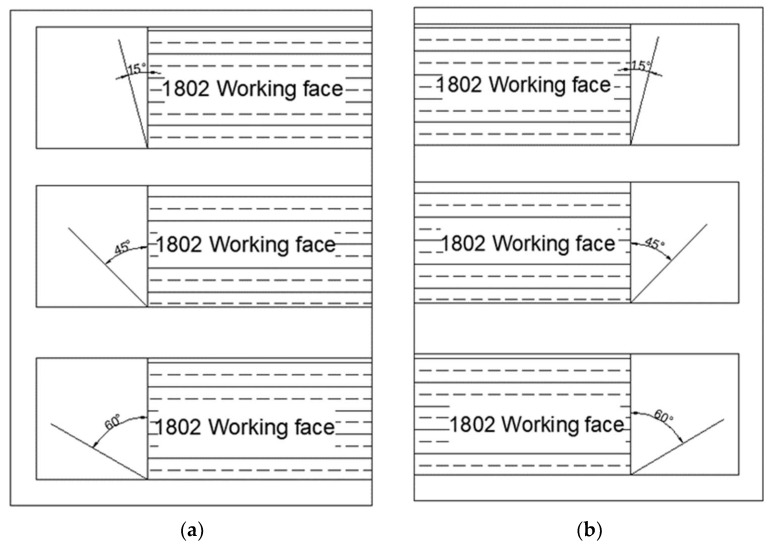
Transient electromagnetic detection angle. (**a**) Return air lane; (**b**) transport lane.

**Figure 10 sensors-24-01771-f010:**
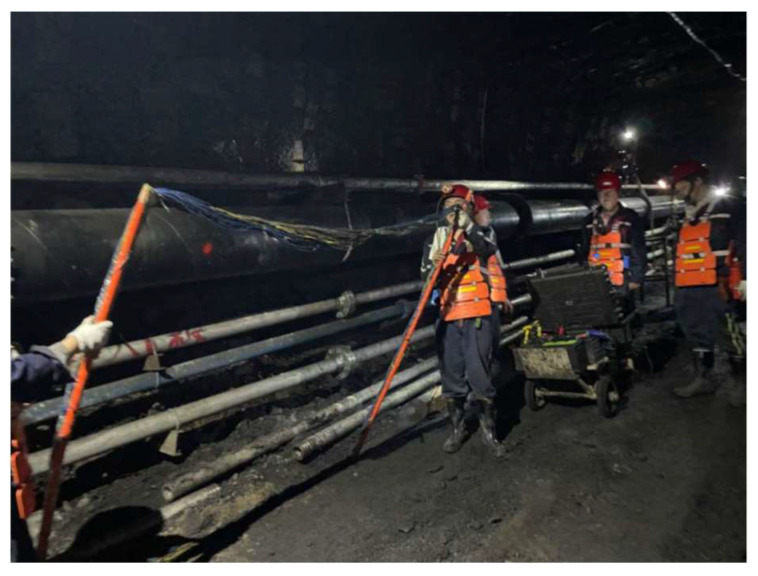
Transient electromagnetic detection site of drilling field 1.

**Figure 11 sensors-24-01771-f011:**
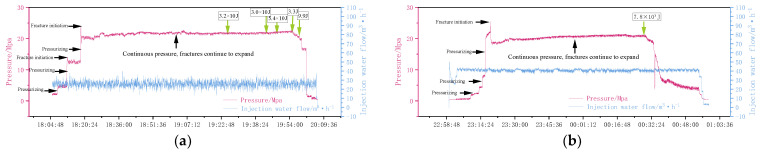
Fracturing pressure–injection water flow curve diagram. (**a**) The fifth stage of borehole No. 1; (**b**) the fifth stage of borehole No. 3.

**Figure 12 sensors-24-01771-f012:**
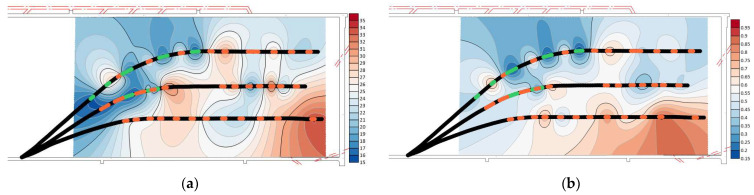
Fracturing pressure cloud map of each fracturing section of boreholes in drilling field 1. (**a**) Fracture initiation pressure (MPa); (**b**) ratio of injection pressure to injection flow in the stable stage (MPa/m^3^).

**Figure 13 sensors-24-01771-f013:**
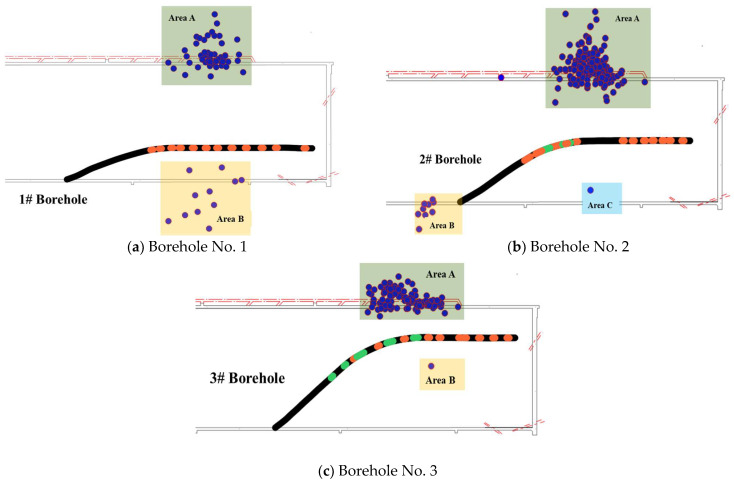
Microseismic distribution map monitored during fracturing at drilling field 1.

**Figure 14 sensors-24-01771-f014:**
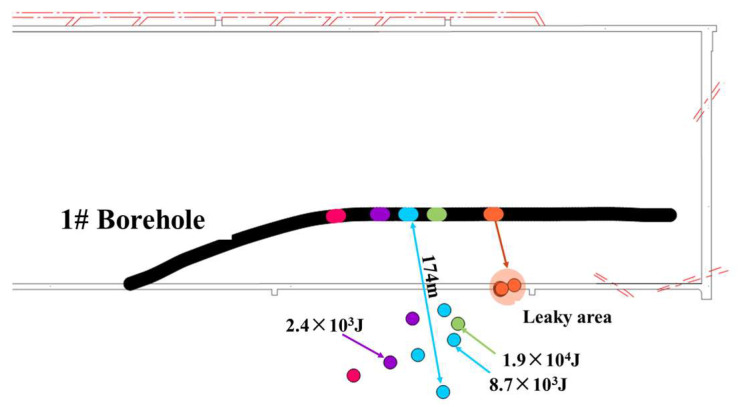
Distribution of hydraulic fracturing microseismic signals in hole No. 1.

**Figure 15 sensors-24-01771-f015:**
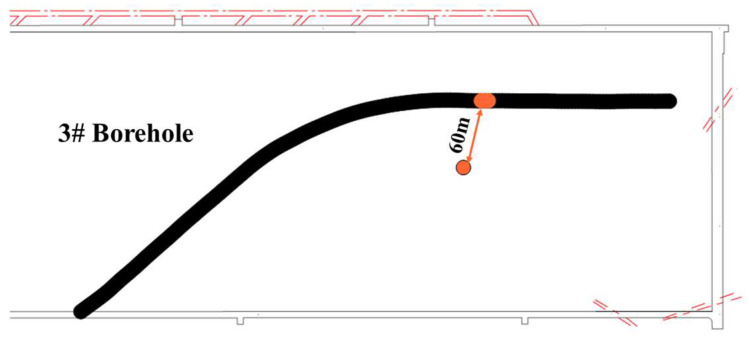
Distribution of hydraulic fracturing microseismic signals in hole No.3.

**Figure 16 sensors-24-01771-f016:**
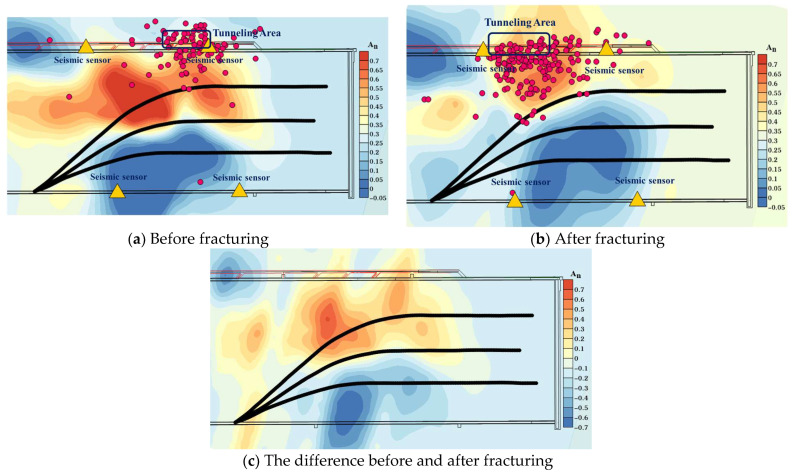
Cloud chart of abnormal wave velocity value in drilling field 1.

**Figure 17 sensors-24-01771-f017:**
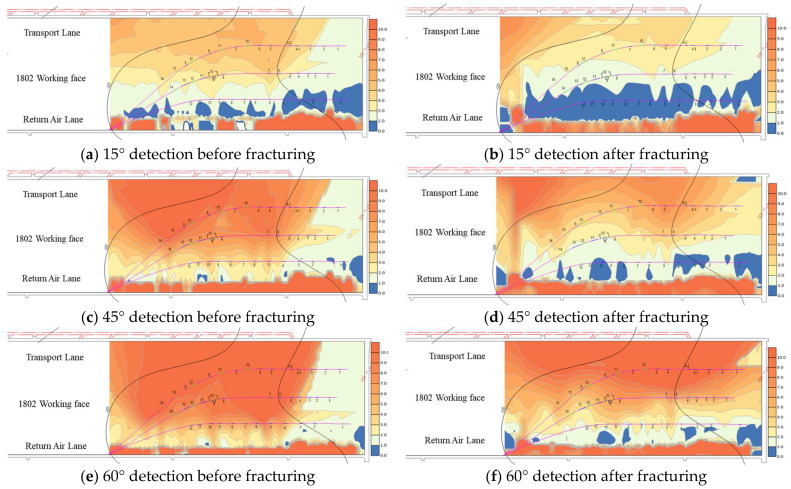
Working face 0 m~530 m apparent resistivity contour profile.

**Figure 18 sensors-24-01771-f018:**
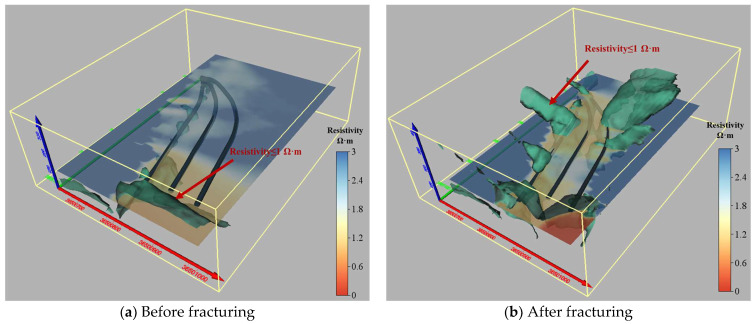
Apparent resistivity distribution along borehole plane and vertical profile.

**Figure 19 sensors-24-01771-f019:**
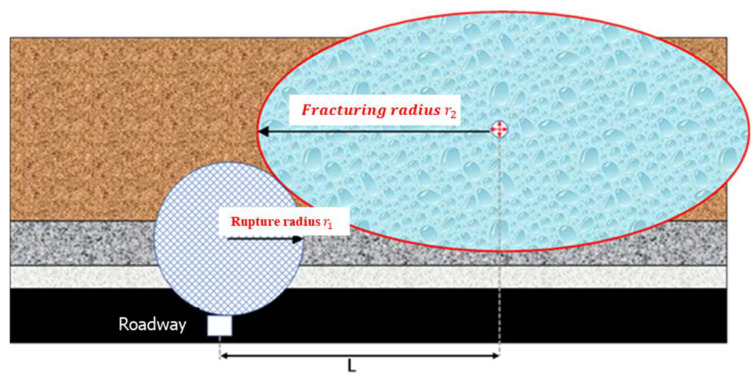
Schematic diagram of fracturing radius calculation.

**Figure 20 sensors-24-01771-f020:**
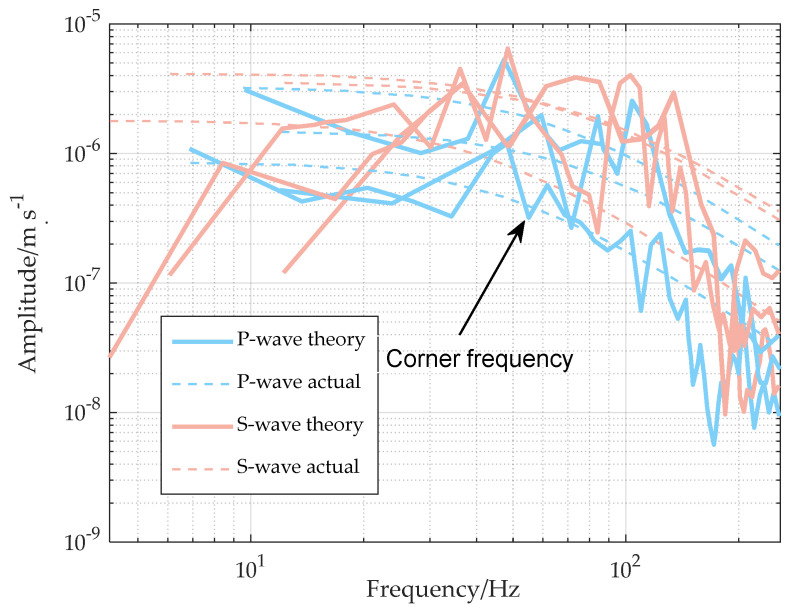
Fracturing disturbance microseismic signal spectrum.

**Figure 21 sensors-24-01771-f021:**
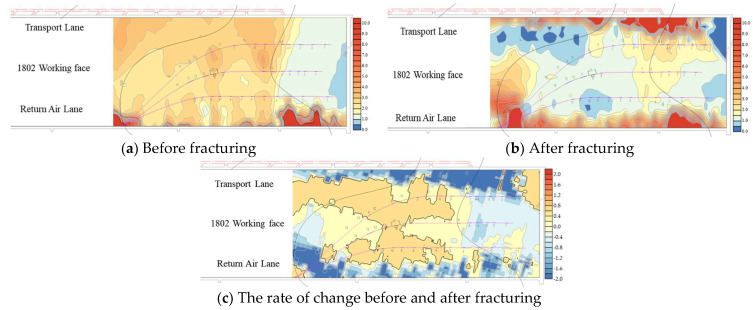
Apparent resistivity distribution along the horizontal section of the boreholes.

**Figure 22 sensors-24-01771-f022:**
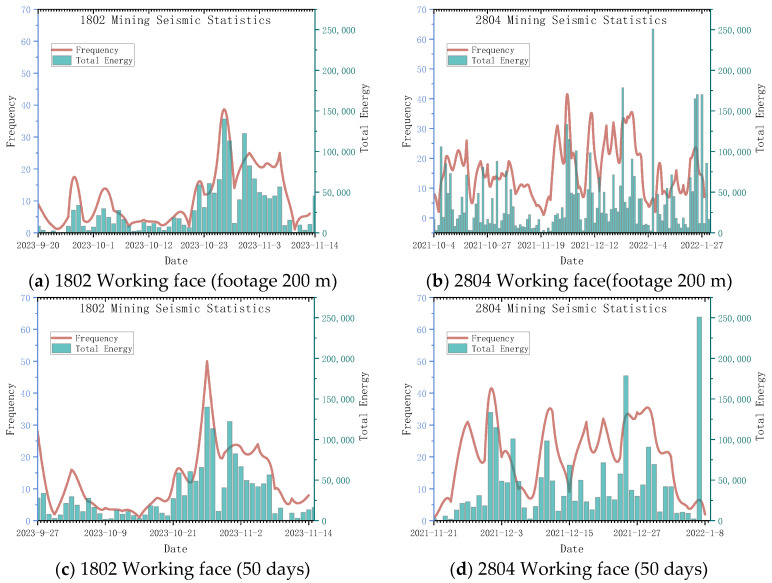
Statistics of initial mining microseismic data of working face.

**Figure 23 sensors-24-01771-f023:**
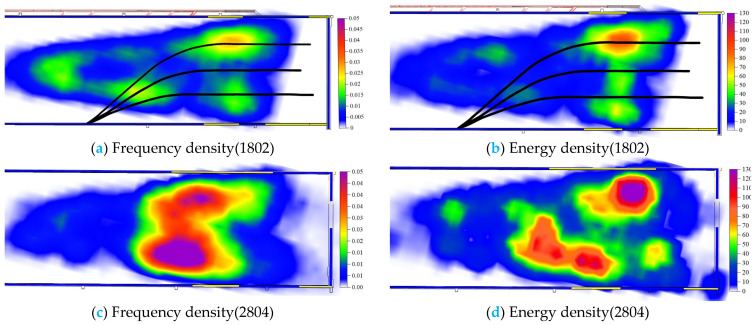
Cloud map of spatial evolution of microseismic frequency energy in 1802 and 2804 working face.

**Table 1 sensors-24-01771-t001:** Drilling design fracturing parameters of drilling field 1.

Borehole	Design Hole Depth/m	Aperture/mm	Number of Stages
1#	528	120	12
2#	519	120	12
3#	560	120	12

**Table 2 sensors-24-01771-t002:** Length of fracturing section of each borehole in drilling field 1.

Stage of Fracture	Fracturing Range of Hole No. 1/m	Fracturing Length of Hole No. 1/m	Fracturing Range of Hole No. 2/m	Fracturing Length of Hole No. 2/m	Fracturing Range of Hole No. 3/m	Fracturing Length of Hole No. 3/m
1	512.27–519.85	7.58	503.22–510.8	7.58	555.99–562.57	6.58
2	452.27–459.85	7.58	476.22–483.8	7.58	525.99–532.57	6.58
3	422.27–429.85	7.58	461.22–468.8	7.58	495.99–502.57	6.58
4	389.27–396.85	7.58	440.22–447.8	7.58	453.99–472.57	6.58
5	359.27–366.85	7.58	422.22–429.8	7.58	410.99–417.57	6.58
6	329.27–336.85	7.58	398.22–405.8	7.58	385.99–392.57	6.58
7	305.27–312.85	7.58	377.22–384.8	7.58	334.99–341.57	6.58
8	278.27–285.85	7.58	278.22–285.8	7.58	278.99–285.57	6.58
9	251.27–258.85	7.58	257.22–264.8	7.58	221.99–228.57	6.58
10	233.27–240.85	7.58	251.22–270.8	19.58	356.99–370.57	13.58
11	209.27–216.85	7.58	230.22–246.8	16.58	300.99–314.57	13.58
12	191.27–198.85	7.58	209.22–225.8	16.58	230.99–250.57	19.58
13	/	/	191.22–207.8	16.58	197.99–205.57	7.58
14	/	/	164.22–180.8	16.58	161.99–169.57	7.58

**Table 3 sensors-24-01771-t003:** Basic parameters of microseismic sensors.

	Underground Sensors (SOS)	Ground Sensorss (ET-GSY)
Range	0.625 mm/s (small range); 0.5 m/s (Medium rang); 1 m/s (wide range)	±1 g/±2 g/±4 g
Type of transmission	Current mode	Built-in three-way EpiSensor force-balanced accelerometer
Bandwidth	0.1~600 Hz	DC~200 Hz
Working temperature	−5 °C~50 °C	−30~+70 °C
Supply district	18~42 V	9~28 VDC

**Table 4 sensors-24-01771-t004:** Relationship between wave velocity outliers and stress concentration probability.

Stress Concentration	Wave Velocity Outlier $A_{n}$	Probability of Stress Concentration
Weak grade	0~0.15	<0.6
Medium grade	0.15~0.25	0.6~1.4
Strong grade	>0.25	>1.4

## Data Availability

The seismic data and apparent resistivity data used in this work are from Hetaoyu Coal Mine, and they are confidential.

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
