# Peer review of "A Combined Method of Seismic Monitoring and Transient Electromagnetic Detection for the Evaluation of Hydraulic Fracturing Effect in Coal Burst Prevention"

_sensors, 2024, doi:10.3390/s24061771_

Round 1

Reviewer 1 Report

Comments and Suggestions for Authors

Author Response

Thanks so much for taking your time to review this manuscript. We gratefully appreciate the editors and all reviewers for their positive and constructive comments. These comments are valuable and helpful for revising and improving the manuscript entitled “A combined method of seismic monitoring and transient electromagnetic detection for the evaluation of hydraulic fracturing effect in coal burst prevention (Manuscript ID: sensors-2866023), as well as the important guiding significance to our researches.

We have studied comments carefully and have made correction which we hope meet with approval. Revised portion have been marked and corrected in the manuscript through the review mode, and the responses to the itemized reviewer's comments are listed in the Revision Report

Reviewer 2 Report

Comments and Suggestions for Authors

This paper introduced a combined underground-ground integrated microseismic monitoring and transient electromagnetic detection method for evaluation hydraulic fracturing effect. It has certain practical application value in engineering. Some comments for this paper:

1. Lines 80-83 already describe the evaluation of the effectiveness of hydraulic fracturing using a joint approach; would lines 84 through 85 be redundant? Perhaps you mean to express that the two sections respectively involve the use of passive CT imaging and localization results?

2. Please identify the units and physical meanings of the color bar in the fig.1.

3. In Figure 16, what physical information does the color represent? What are the units? How do you get  graph a and b? Is it only possible to obtain graph c through equation (2)?

Comments on the Quality of English Language

Minor editing of English language required. I'm a little confused by the presentation of lines 80-85, and would suggest a revision.

Author Response

(The authors gave the same response as above.)
